# REASONING BOOSTS OPINION ALIGNMENT IN LLMS

**Frédéric Berdoz, Yann Billeter, Yann Vonlanthen, Roger Wattenhofer**
ETH Zurich
Switzerland
`{fberdoz, ybilleter, yvonlanthen, wattenhofer}@ethz.ch`

## ABSTRACT

Opinion modeling aims to capture individual or group political preferences, enabling applications such as digital democracies, where models could help shape fairer and more popular policies. Given their versatility, strong generalization capabilities, and demonstrated success across diverse text-to-text applications, large language models (LLMs) are natural candidates for this task. However, due to their statistical nature and limited causal understanding, they tend to produce biased opinions when prompted naively. In this work, we study whether reasoning can improve opinion alignment. Motivated by the recent advancement in mathematical reasoning enabled by reinforcement learning (RL), we train models to produce profile-consistent answers through structured reasoning. We evaluate our approach on three datasets covering U.S., European, and Swiss politics. Results indicate that reasoning enhances opinion modeling and is competitive with strong baselines, but does not fully remove bias, highlighting the need for additional mechanisms to build faithful political digital twins using LLMs. By releasing both our method and datasets[1], we establish a solid baseline to support future research on LLM opinion alignment.

## 1 INTRODUCTION

Could AI give rise to a new kind of democracy, where digital twins vote on our behalf and faithfully reflect our opinions on every issue? Accurate simulations of political behavior offer new opportunities to understand election outcomes and improve policy and democratic processes (Li et al., 2024). However, capturing how diverse individuals reason about political issues remains challenging. While large language models (LLMs) can generate sophisticated political discourse, they often fail to reflect the true diversity of human political viewpoints (Santurkar et al., 2023; Qu & Wang, 2024; Yu et al., 2025; Qi et al., 2024). This raises a fundamental question: how to design agents that reason about politics while faithfully representing human political diversity?

Current approaches to modeling opinions with LLMs predominantly rely on prompting with demographic information and political affiliations, thereby leveraging learned correlations between demographics and opinions. While these prompt-based methods have grown increasingly sophisticated, they consistently fail to capture real opinion distributions and exhibit unstable, inconsistent responses across different prompts and demographic groups (Santurkar et al., 2023; Ball et al., 2025). We instead seek a preference-consistent approach that directly uses an individual's known opinions rather than demographic proxies.

Yet, the aforementioned methods do not lend themselves to simulating individual political preferences. Methods using richer data sources like interview transcripts can achieve strong performance in modeling individual personas (Park et al., 2024), but remain limited by prohibitive data collection costs, as interviewing is impractical at scale. In this paper, we propose using political survey data as an alternative. Large surveys, like the American National Election Studies (ANES), and Voting Advice Applications (VAAs), i.e., online platforms that match voters with parties or candidates, aim to capture how people position themselves on political issues. While surveys lack the narrative richness and textual signal of interviews, they offer a structured representation of political opinions across populations. Despite their prevalence in bias evaluation (Santurkar et al., 2023; Argyle et al.,

---

[1]Code and data: `https://github.com/ETH-DISCO/reasoning-boosts-llm-alignment`

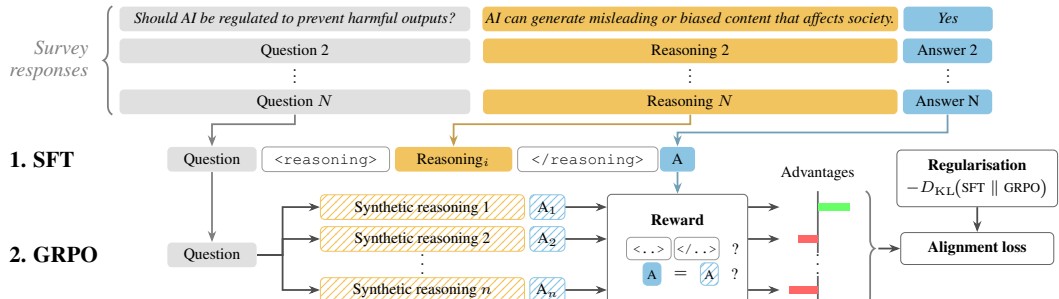

Figure 1: **GRPO for opinion alignment**. We use public opinion surveys to align LLMs with individual preference profiles. First, the LLM is fine-tuned with (synthetic) statements and ground-truth answers to adhere to the reasoning template. After fine-tuning, the model answers in the correct format, but is not fully aligned with opinions. We use GRPO with a reward model that rewards proper formatting and correct answers to further improve reasoning.

2023; Haller et al., 2025) and group-level simulation (Argyle et al., 2023; Cao et al., 2025), surveys have not yet been widely adopted for training individual-level political agents.

RL has recently driven major gains in reasoning, with models like o3/o4-mini (OpenAI, 2025) and DeepSeek-R1 (Guo et al., 2025) setting new marks on various benchmarks. Motivated by these results, we use RL, in particular GRPO (Shao et al., 2024), to train agents on survey data, treating opinion formation as a reasoning problem: the agent writes a short rationale and a final choice, and is rewarded when the choice matches the respondent's answer, teaching it to reason toward survey-consistent positions on unseen questions.

We summarize our contributions as follows:

- **Reinforcement learning for reasoning-based opinion alignment.** We introduce a method that encourages explicit reasoning through reinforcement learning (GRPO) to improve opinion alignment from survey responses.
- **Benchmarking on real-world political data.** We evaluate our approach on three datasets of authentic political opinions: German parties, U.S. voters, and Swiss candidates, and release them as a benchmark to foster future research. Our results show that reasoning consistently improves opinion alignment.
- **Analyzing ideological effects.** We investigate how political ideology influences performance by inverting the survey answers of individuals with known positions, revealing systematic variations in alignment quality across the political spectrum.

## 2 RELATED WORK

Modeling political opinions requires LLMs to adopt specific political personas and answer from these particular viewpoints. Prior work has explored *demographic personas* (representing groups with shared traits like occupation or ethnicity), *character personas* (well-known individuals), and *individualized personas* (digital profiles based on personalized data) (Chen et al., 2024; Tseng et al., 2024). Our focus on modeling individuals makes individualized personas most relevant, though the literature has predominantly employed demographic approaches, with individualized methods represented by only a handful of studies.

**Modeling political opinions with demographic personas.** Political opinion modeling has primarily used demographic personas through various prompting strategies. Simple approaches use basic political affiliation (e.g., "You are a democrat. What is your opinion on...") (Santurkar et al., 2023), while more complex methods add demographic details like age, education, and location (Argyle et al., 2023; Santurkar et al., 2023; Hwang et al., 2023; Sun et al., 2024; Wang et al., 2025). Recent work has also explored richer contexts, including prompts with demonstrations from belief networks, where models receive related political positions to guide their answer (Chuang et al.,

Table 1: Answers obtained from Llama 3.1 8B trained on an individual in the Right group in *ANES*. Before training, Llama disagreed; after training, it agreed. Interestingly, the reasoning trace still appeals to similar arguments ("equal opportunity") but frames them differently. Additional examples are provided in Appendix B.

| **Question:** |
| --- |
| Do you agree that it is not really that big a problem if some people have more of a chance in life than others? |

| **Before** (prompting on base model) |
| --- |
| No. I don't think that's right. Everyone deserves an equal shot at succeeding in life, no matter where they're from or who their parents are. If some people have more opportunities and resources, it's not fair to others. We should be working to level the playing field and give everyone a fair chance to succeed. |

| **After** (`SFT+GRPO`) |
| --- |
| `<reasoning>` Some inequality in wealth and opportunity is inevitable and indeed necessary to provide incentives for hard work and innovation – the issue is not eliminating all differences but rather ensuring that everyone has access to basic necessities and opportunities to improve themselves – other countries with significant equality have still struggled with poverty and stagnation.`</reasoning>` `<answer>`A) Yes`</answer>` |

2024). Gudiño et al. (2024) combine demographics with demonstrations, while Stammbach et al. (2024) integrate prompting with odds ratio preference optimization (ORPO) (Hong et al., 2024) to better align outputs with target positions. Despite this methodological diversity, demographic approaches face several limitations.

**Limitations of demographic approaches.** Santurkar et al. (2023) identify three critical issues of prompt-based demographic simulation that undermine authentic political representation: *representativeness*, where LLMs' default opinion distributions fail to reflect true population distributions; *steerability*, where demographic prompting provides insufficient control over model outputs; and *consistency*, where political biases vary unpredictably across different topics and contexts. These limitations manifest as systematic biases, with accuracy varying unpredictably across demographic groups (Argyle et al., 2023; Santurkar et al., 2023). While models can approximate overall opinion distributions, they fail to capture natural variance within demographic subgroups and remain sensitive to prompt variations (Sun et al., 2024; Ball et al., 2025).

**Moving beyond demographics.** In contrast to the extensive literature on demographic prompting, few studies have explored representing individual opinion profiles. Park et al. (2024) use interview transcripts to achieve strong accuracy on public opinion surveys, outperforming demographic baselines, but this approach is limited by transcript availability. Haller et al. (2025) propose an in-context learning method, where the model is prompted with respondents' question-answer pairs and then asked to predict the respondents' answers to a set of test questions. However, their focus was on bias identification rather than opinion modeling.

**Training LLMs to reason.** Encouraging LLMs to produce reasoning steps before answering significantly increases accuracy, with approaches like Chain-of-Thought prompting (Wei et al., 2022) or simple prompts like "Let's think step by step" (Kojima et al., 2022). Recent advances have shifted from prompting-based reasoning to training models to generate high-quality reasoning traces through search algorithms or reinforcement learning (Xu et al., 2025). One particular method is Group-relative policy optimization (GRPO) (Shao et al., 2024). GRPO is a variant of Proximal policy optimization (PPO) (Schulman et al., 2017), that uses a group-relative reward normalization rather than learning a value function to estimate advantages. For each prompt, GRPO samples a group of outputs and normalizes rewards by subtracting the group mean and dividing by the group standard deviation. GRPO has proven particularly effective for mathematical reasoning tasks, achieving state-of-the-art results on benchmarks (Guo et al., 2025).

**Reasoning for opinion alignment.** Yu et al. (2025) propose a persona-based chain-of-thought-inspired prompting framework, which combines demographic, ideological, and temporal factors calibrated using ANES. The authors use this framework to simulate the voting behavior of individual respondents and sum these responses to make aggregate predictions. Unlike our proposed method, they do not use reinforcement learning. Moreover, we focus on learning general preference profiles,

Table 2: Dataset overview. Train/test counts refer to questions per unit (candidate/party/respondent).

| Setting | Unit | #Units | Labels | Train Q | Test Q |
|---|---|---|---|---|---|
| *smartvote* (CH) | Candidate | 18 | {Yes, No} | 48 | 12 (topic-strat.) |
| *WoM* (DE) | Party | 6 | {Yes, Neutral, No} | see Table 5 | 30 (EU&I) |
| *ANES* 2020 (US) | Respondent | 21 | {Yes, Neutral, No} | 67 | 12 (random) |

while Yu et al. (2025) target voting predictions. While our method could theoretically be applied to the same problem, the computational cost of modeling a representative sample size is currently prohibitive.

## 3 METHOD

We study individual-level opinion alignment from survey data. Consider policy questions such as "Should the government increase funding for renewable energy?" with answer stances like $Y = \{\texttt{No}, \texttt{Neutral}, \texttt{Yes}\}$. Given any policy question $q$ and persona $p$ (a respondent or political party), our goal is to learn the stance of $p$ on $q$.

### 3.1 POLITICAL REASONING WITH GRPO

We employ GRPO (Shao et al., 2024) to train agents that generate reasoning traces before providing answers in the following fixed schema:

$$\texttt{<reasoning>[justification text]</reasoning><answer>[stance]</answer>}. \quad (1)$$

Our training data consists of survey responses, $\mathcal{D} = \{(p_j, q_i, y_{ij}^*)\}$, where $y_{ij}^*$ is the observed stance of persona $p_j$ on question $q_i$. The key challenge is that $\mathcal{D}$ contains no reasoning traces, only final stances. Therefore, our model must learn to generate reasoning that both adheres to the required format above and produces answers that match the ground-truth stances. To achieve this, we use a composite reward function to evaluate and reward each generation along multiple dimensions:

*Format Reward* ($R_{\text{format}}$): We enforce a structured output format by rewarding correctly placed tags in the schema described in (1), which yields a score of 1 for each of the four tags (maximum of 4).

*Length Reward* ($R_{\text{length}}$): Let $L$ be the token length of the reasoning trace $x_i$ and $L^*$ the desired length. We define a symmetric penalty: $R_{\text{length}} = -|L - L^*|$, which yields a maximum reward of 0 only when the trace length equals $L^*$.

*Correctness Reward* ($R_{\text{correct}}$): The primary reward signal comes from matching the survey response: $R_{\text{correct}} = \mathbb{1}[y_i = y_i^*]$, where $y_i$ is the model's predicted answer and $y_i^*$ is the ground-truth survey response. The total reward is computed as: $R(r_i, y_i^*) = \alpha_1 R_{\text{format}} + \alpha_2 R_{\text{length}} + \alpha_3 R_{\text{correct}}$, with $\alpha_1 = 0.25, \alpha_2 = 0.01, \alpha_3 = 1$.

We do not use an explicit persona representation, and include only a country label in the system prompt. The model is aligned with the individuals solely through correctly answering questions.

**Optional Initialization via Supervised Fine-tuning.** To accelerate convergence and improve initial reasoning quality, we optionally initialize the model through supervised fine-tuning (SFT) before GRPO training. Following (Shao et al., 2024), we construct a dataset of chain-of-thought demonstrations, where each example follows the exact format (1) incentivized by our reward design. The SFT stage serves as a warm start that: (i) reduces the burden on $R_{\text{format}}$ during GRPO by pre-training the model to use the correct output structure, and (ii) provides a reasonable initialization for political reasoning.

Table 3: **Reasoning requires large datasets to be effective**. Mean macro-F1 (%) with standard deviations over 8 stochastic runs at $T{=}1.0$. For each run, we compute per-unit macro-F1 and then average across units. We report accuracy scores and statistical significance in Appendix E.1. Scores on *smartvote* are considerably better than on *ANES* and *WoM*. This is likely due to the binary nature of *smartvote* and the absence of a `Neutral` option (see Section 4.6). Performance on *WoM* is better than on *ANES*, likely because of the substantially bigger number of training samples (see also Appendix E.4).

| Base model | Method | Dataset | | |
|---|---|---|---|---|
| | | *smartvote* | *WoM* | *ANES* |
| Untrained baselines | `random` | 50.00 | 33.33 | 33.33 |
| | `majority` | 37.43 | 27.44 | 22.98 |
| Llama 3.1 8B | `icl` | $55.97 \pm 5.66$ | $28.17 \pm 0.02$ | $23.20 \pm 1.79$ |
| | `ORPO` | $43.53 \pm 2.81$ | $43.29 \pm 5.16$ | $34.84 \pm 5.41$ |
| | `SFT` | $63.44 \pm 1.34$ | $48.95 \pm 3.56$ | $\mathbf{42.77 \pm 1.23}$ |
| | `GRPO` | $55.14 \pm 1.25$ | $37.29 \pm 2.06$ | $34.55 \pm 1.32$ |
| | `SFT+GRPO` | $\mathbf{66.88 \pm 2.18}$ | $\mathbf{52.53 \pm 4.05}$ | $40.66 \pm 0.91$ |
| Qwen3 8B | `icl` | $60.48 \pm 4.14$ | $26.19 \pm 0.35$ | $23.20 \pm 1.79$ |
| | `ORPO` | $23.87 \pm 2.07$ | $25.25 \pm 3.58$ | $26.95 \pm 1.36$ |
| | `SFT` | $61.08 \pm 2.74$ | $42.91 \pm 3.44$ | $35.14 \pm 1.70$ |
| | `GRPO` | $60.64 \pm 2.01$ | $31.42 \pm 1.46$ | $31.47 \pm 1.06$ |
| | `SFT+GRPO` | $\mathbf{65.11 \pm 3.33}$ | $\mathbf{49.38 \pm 1.93}$ | $\mathbf{38.44 \pm 0.40}$ |
| Magistral 24B | `icl` | $66.16 \pm 0.42$ | $26.19 \pm 0.35$ | $19.23 \pm 2.91$ |
| | `ORPO` | $23.31 \pm 2.11$ | $24.73 \pm 3.29$ | $24.25 \pm 2.28$ |
| | `SFT` | $67.63 \pm 1.91$ | $51.86 \pm 2.58$ | $39.15 \pm 0.66$ |
| | `GRPO` | $60.56 \pm 1.93$ | $51.00 \pm 3.10$ | $43.79 \pm 1.06$ |
| | `SFT+GRPO` | $\mathbf{70.73 \pm 2.21}$ | $\mathbf{53.21 \pm 3.19}$ | $\mathbf{45.43 \pm 1.11}$ |

## 4 EXPERIMENTS

### 4.1 DATASETS

For our experiments, we use three datasets from different countries and political systems. The dataset creation process is described in the following paragraphs. We release all data for reproducibility and future research. An overview of the dataset is provided in Table 2.

**German party positions.** For training, we use the official *Wahl-O-Mat* (WoM) dataset (version 26 March 2025) aggregating party positions across federal, state, and European elections (2021–2025). We focus on six major parties (CDU/CSU, SPD, Grüne, FDP, Die Linke, AfD). Each item records `agree`/`neutral`/`disagree` and an explanatory comment. We map to {`Yes`, `Neutral`, `No`}. For testing, we use *EU&I* (2024) (Chalkidis, 2024) as an out-of-domain questionnaire (30 items), recoded to the same three labels. Per-party training counts appear in Appendix A.

**American National Election Studies.** From ANES 2020 Time Series (American National Election Studies, 2021), we extract 79 policy items (excluding demographics, knowledge, candidate evaluations, and thermometers). Because response formats are heterogeneous, we recode each item to {`Yes`, `Neutral`, `No`} using two schemes detailed in Appendix D: a *conservative* mapping that treats only clearly positive/negative options as agreement/disagreement, and an *aggressive* mapping that collapses most positive responses to `Yes`. Unless otherwise stated, results use the conservative scheme. Robustness to recoding is analyzed in Appendix E.2. We split items at random into 67 train and 12 test questions. To analyze ideological variation, we sample respondents based on how they place themselves on *ANES*' 7-point liberal–conservative self-placement question. We included 3 respondents per answer option, for a total of 21 respondents.

**Swiss candidates.** We use the English questionnaire for the 2023 Swiss national elections (75 items, of which 60 are policy questions across 12 topics). Responses are on a four-point Likert scale and are collapsed to {`Yes`, `No`} by mapping `Yes`/`Rather yes` to `Yes` and `Rather no`/`No` to `No`. We sample three candidates from each of the six parties with at least two National Council seats (18 candidates total). See Appendix A for additional information on these parties. We perform a topic-stratified split: one question per topic held out for test (12), the remaining 48 for training.

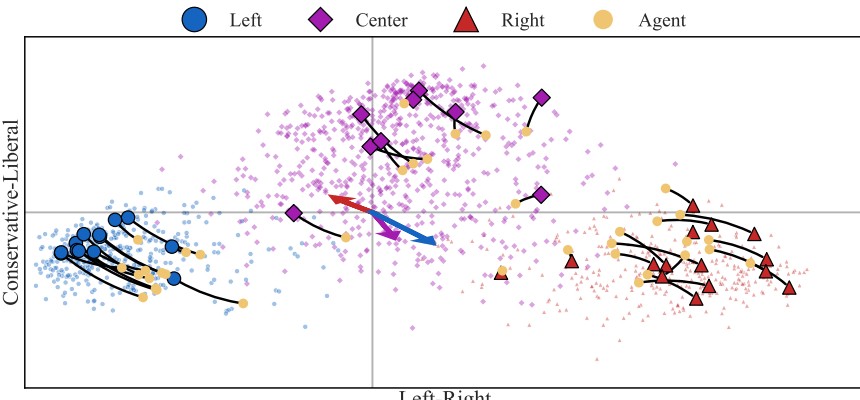

Figure 2: **Agents are more centrist and conservative**. First two dimensions of the principal component analysis of all candidates (small dots) standing for election in the 2023 Swiss national elections (*smartvote*). The x- and y-axes correspond to the left-right and conservative-liberal spectra, respectively. The shifts between the positions of the candidates (big dots) included in the *smartvote* dataset and their agents (gold) are depicted by black lines. Unlike results in the literature which indicate a left-libertarian bias (Exler et al., 2025; Hartmann et al., 2023; Rozado, 2024), we observe that our agents are shifted towards the (center-)right. The average distortions between ground-truth and agent position for each group (big arrows) show an overall trend towards more conservative (negative y) and no clear left-right bias. Implementation details are given in Appendix E.5.

**Political Spectrum Groups.** For clarity and comparability across political contexts, we assign parties (*smartvote*, *Wahl-o-Mat*) and ideologies (*ANES*) to three broad political groups: Left, Center, Right. The assignments are detailed in Appendix A.

## 4.2 EXPERIMENTAL SETUP

We evaluate our approach on different models and against strong baselines. The design targets two questions: (i) how performance scales with model size, and (ii) whether reasoning-pretrained backbones confer advantages over non-reasoning peers. We train one model per unit (candidate/party/respondent). Further training details are given in Appendix F.

**Methods** We report results for three configurations: GRPO on the survey questions and answers, as well as SFT+GRPO. For *smartvote* and *ANES*, we generate synthetic arguments supporting both positive and negative stances for each policy question using Llama 3.1 70B (Grattafiori et al., 2024) and use those for SFT. The prompting strategy used to produce these synthetic arguments is detailed in Appendix A.4.

**Model backbones..** We evaluate across three open-weight backbones to study (i) the effect of model scale and (ii) the impact of prior reasoning pre-training: **Llama3.1 8B** (Grattafiori et al., 2024), **Qwen3 8B** Yang et al. (2025), and **Magistral 24B** Rastogi et al. (2025). Qwen3 and Magistral are pre-trained to reason, allowing us to compare reasoning-pretrained backbones against a non-reasoning counterpart (Llama3.1) at comparable scales. We use 4-bit quantization for all models across all experiments that involve training.

**Baselines** We report results for two naive baselines. random selects one of the answers uniformly at random. majority answers all questions with the most frequently chosen answer option in that unit's train set. Additionally, we also compare to an in-context learning baseline icl. We follow the methodology described in (Haller et al., 2025), where the LLM is prompted with question-answer pairs and then asked to answer another unseen question. To do so, we report results for the setting where the model receives all the training questions from the same topic area as the test question (or a random subset when limited by context length). Finally, we compare to ORPO Hong et al. (2024) using the same hyperparameters as Stammbach et al. (2024).

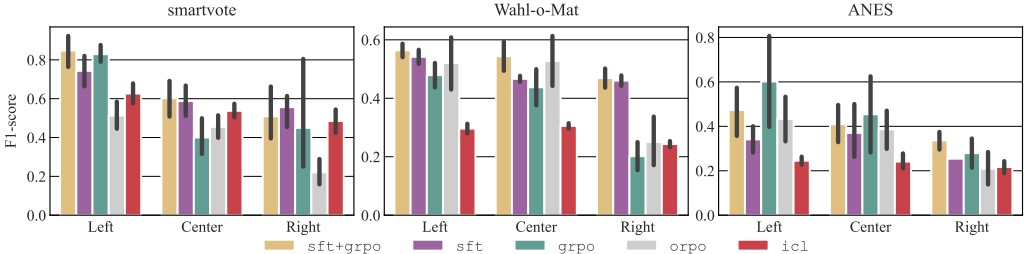

Figure 3: **Not all political positions are equally learnable**. F1 scores reveal that while `SFT+GRPO` typically works best, every training method underperforms on center and right-leaning groups. Error bars show variance within groups. These disparities may stem from biases baked into Llama 3.1 8B or from inherent differences in how well various political preferences can be learned from survey data. Either way, the results demonstrate that ideology impacts the learnability of political preferences. A detailed figure with the original parties and ideology groups is presented in Appendix C.

## 4.3 RESULTS AND ANALYSIS

Table 3 reports macro-F1 averages for all datasets. `SFT+GRPO` outperforms naive and `SFT` baselines across datasets, except for Llama 3.1 8B on *ANES*. Performance varies by dataset: scores peak on *smartvote* (70.73), while *ANES* is hardest (45.43). `SFT+GRPO` performs comparably to or better than `icl`-baselines on *ANES* and *smartvote*, and outperforms them on *WoM*. Using base models with native reasoning support does not consistently improve performance, as Llama 3.1 8B outperforms Qwen3 8B on *WoM* and *ANES* while Magistral 24B performs best overall. `GRPO` alone underperforms `SFT+GRPO`, suggesting supervised warm start improves training dynamics. We observe lower performance on datasets with `Neutral` classes, likely because `Neutral` aggregates multiple behaviors (uncertainty, social desirability, strategic non-commitment), making it difficult to learn. We further investigate ideology impact in Section 4.5 and `Neutral`'s role in Section 4.6. Results without `Neutral` appear in Appendix E.1.

## 4.4 HOW DO AGENTS REASON?

In Table 1, we present an example of reasoning traces. Generally, agents succeed in making coherent arguments for or against a proposed measure that imply the final answer. We identify two failure modes: First, arguments sometimes contain illogical or hallucinated elements. Second, an agent may argue in line with an individual's opinion on one topic, but then fail to do so on other topics. Additional traces and examples of failures are provided in Appendix B.

## 4.5 POLITICAL IDEOLOGY MATTERS

In Fig. 3, we report F1 by ideology group for Llama 3.1 8B (`SFT+GRPO`). On *smartvote*, we observe a clear performance disparity between the Left group and the Center and Right groups. *Wahl-o-Mat* and *ANES* show similar patterns, albeit weaker: On *Wahl-o-Mat*, the Left and Center groups perform roughly equally well, while performance is generally lower on *ANES*. For the Right group in *smartvote* and *ANES*, `SFT` also outperforms all other methods on average, further suggesting difficulties with right-wing preference profiles. These disparities may reflect known left-leaning tendencies in general-purpose LLMs (Hartmann et al., 2023; Exler et al., 2025) or indicate that Right profiles are intrinsically harder to learn from survey signals. The relatively strong Right performance on *WoM* may be aided by larger training sets (cf. Appendix E.4).

## 4.6 NEUTRAL REASONING POSES A PARTICULAR CHALLENGE

As shown in Fig. 4 (top right), recall is worst on the `Neutral` class for all individuals in the *ANES* dataset. Furthermore, as can be seen from the regression in Fig. 4 (left column), individuals in the Right group respond with `Neutral` the most (higher neutral base rate). Clearly, performance for these individuals suffers the most from the difficulties in predicting `Neutral`. Removing all

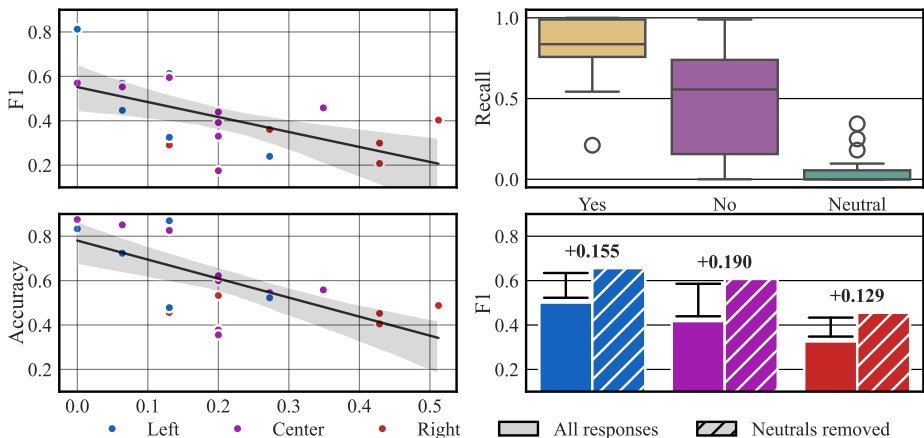

Figure 4: **Learning neutral stances is hard**. On *ANES*, there is a strong correlation between predictive performance and the neutral base rate on the test set. Individuals in the Right group, and to some extent in the Center group, tend to answer questions with `Neutral` more often. Both in terms of F1 (top left) and accuracy (bottom left), we observe a negative correlation significant at the 5%-level. Regression details in Appendix E.3. Top right: Recall by class on *ANES*. Averaged over all individuals, the model struggles the most with predicting the `Neutral` class. Bottom right: Performance in terms of F1 score on *ANES* before (solid) and after (shaded) removing `Neutral` instances. All groups improve, and the gap between Left and Center becomes narrow. However, the difference between Left and Right remains. While the large number of `Neutral`s likely depressed performance on the other two classes during training, this could also suggest that the performance disparity is affected by other factors, such as model bias or artifacts from recoding *ANES*. All results were obtained from Llama 3.1 8B trained with `SFT+GRPO`.

`Neutral` instances and recomputing the F1 scores leads to a considerable improvement in performance, as shown in Fig. 4 (bottom right), but does not rectify the gap between the Left and Right group. `SFT+GRPO` also improves relative to the `icl`-baselines when `Neutral` is removed (see Appendix E.1). We repeat this analysis under the alternative recoding scheme mentioned in Section 4.1 and further described in Appendix D, and observe the same negative correlation between predictive performance and `Neutral` base rate, but almost identical performance between the Center and Right groups, indicating that this could, to some extent, be an artifact of recoding *ANES*. Details are reported in Appendix E.2.

## 4.7 Whom do agents represent?

To better understand the agents' representativity in semantic terms, we analyze their positions relative to the ground-truth positions of their corresponding human individuals in a two-dimensional model of politics. We perform this on the *smartvote* dataset, for which the interpretation of the principal components is documented (Germann et al., 2015). Figure 2 presents the positions of the individuals included in the *smartvote* dataset in the space spanned by the first two principal components obtained from the full set of candidates in the 2023 Swiss national elections. Agent positions (gold) were then projected into this space. Not all agents represent their respective candidates (big markers) perfectly. However, despite documented left-libertarian biases of most LLMs (Exler et al., 2025; Hartmann et al., 2023; Rozado, 2024), we do not observe an overall left shift. Instead, the agents' positions are shifted towards the center-right of the left-right ($x$-axis) and the center of the conservative-liberal spectrum ($y$-axis). This is further supported by the group-averaged distances between humans and agents shown in the center of Fig. 2: Right individuals become more left-wing, while Left and Center individuals become more right-wing and conservative.

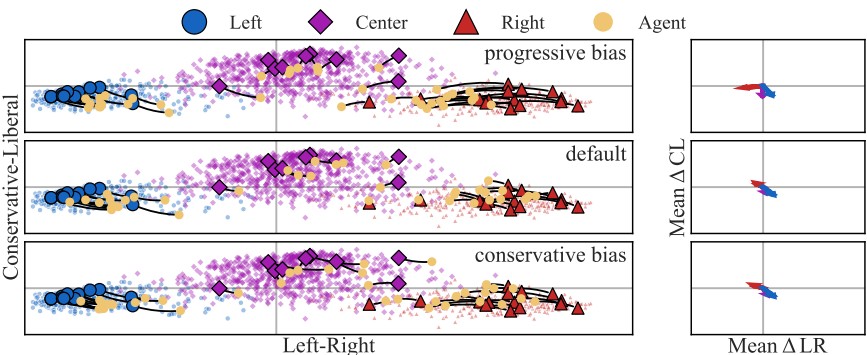

Figure 5: **Biased SFT data consistently impairs the performance of underrepresented groups.** F1-scores for Llama trained with `SFT+GRPO` on differently biased datasets. Left: Data with a progressive bias strongly impairs the Right group, without consistently benefiting the Left group. Middle: Performance on the default dataset. Right: Data with a conservative bias decreases the performance of the Left group, while showing no consistent improvement for the Right group. These asymmetric effects suggest that ideological bias in SFT data primarily harms the underrepresented perspective rather than systematically improving performance on the overrepresented one.

Figure 6: **Biased SFT data primarily shifts the agents of underrepresented groups.** Left: Candidates and their agents in the *smartvote* PCA space. Right: Group-mean displacement vectors of the agents relative to their respective candidates. Similar to the F1 scores presented in Fig. 5, biased data mostly affects the underrepresented group, and Right candidates are again affected more strongly than Left candidates. Notably, conservative-biased and default SFT data produce similar displacement patterns, suggesting the centerward shift in the default data may originate from the base model's alignment rather than the SFT data.

## 4.8 IMPACT OF SFT DATA

To investigate the impact of biases in the SFT data, we generated two additional sets of arguments using Llama 3.3 70B, prompting it to produce progressive and conservative arguments, respectively (details in Appendix A.4). Figure 5 presents the impact in terms of F1 scores. Progressive bias strongly impairs Right candidates while showing inconsistent effects on Left candidates; conservative bias similarly impairs Left candidates without consistently benefiting Right candidates. This suggests that ideological bias in SFT data primarily harms underrepresented perspectives rather than systematically improving overrepresented ones. Notably, Right candidates are impaired more severely by adverse bias than Left candidates. Figure 6 shows how these same biases affect agent positions in PCA space. Consistent with the F1 results, biased data primarily shifts underrepresented groups, with Right candidates again affected more strongly than Left candidates. Interestingly, the positional shifts under conservative-biased and default SFT data are highly similar. This suggests that the centerward shift observed in Fig. 2 may be driven primarily by the alignment of the agent's base model rather than bias in the SFT data itself.

## 4.9 PERFORMANCE ON SYNTHETIC POSITIONS

To assess how individual preferences affect an associated agent's performance, we create a counterfactual by inverting every *smartvote* answer and training Llama-3.1-8B on the resulting dataset. Inversion roughly corresponds to a 180° rotation about the origin in Fig. 2 (see Appendix E.6), con-

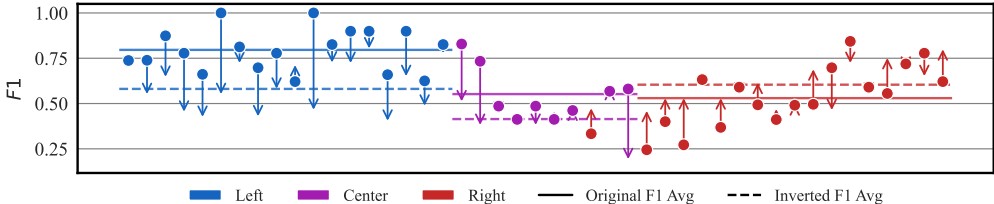

Figure 7: **Learning Right and Center positions is harder than Left ones.** Points depict F1 scores on the original data, with individuals ordered along the x-axis by their first component in the *smartvote* PCA space. Note that the left-most Right candidate is indeed correctly placed (see Fig. 6). Arrows show the difference in F1 scores after all answers in the train and test sets have been flipped. Left candidates are worse after flipping, both individually and on average. Right candidates improve, but still perform worse on average than Left candidates. As the disparity is not removed by converting Left to Right, this could suggest that Right positions are more difficult to learn.

verting, for example, right-wing positions into left-wing ones. For each politician, we compute the F1 difference before and after inversion, as shown in Fig. 7. Members of the Right group benefit from inversion, while the Left and Center groups perform worse on average. Yet even after inversion, the Right does not reach the original F1 of the Left, while the Left drops to roughly the Right's original level. If the effect were purely due to the model's original political bias, flipping labels should have made right-leaning candidates nearly as easy to predict as left-leaning ones. The fact that this does not occur suggests that left-leaning preference profiles may be intrinsically easier to model, a possibility that merits further investigation.

## 5 DISCUSSION

**Limitations.** The learning of human opinions is inherently limited by noise, as even the same respondent might answer a question differently on different days. Hence, consistently achieving an F1-score close to $1.0$ is unlikely. The analyses in this paper cover only a small number of individuals or parties, mainly because the method is computationally intensive (i.e., training one model per profile). Data are another constraint: suitable surveys are scarce. Those that exist often have too few items to train individualized agents, or focus on generic themes rather than concrete issues and priorities. Question sets also rarely align across countries, which limits comparability.

**Future work.** The observed disparities between political groups warrant further investigation into bias mitigation. Further research should aim to replace per-agent training with a single model that conditions on compact persona representations (e.g., see (Ning et al., 2024)), reducing computational cost and improving data utilization. Real-world deployments will require confidence estimates to gauge answer reliability, achievable by extending the reasoning scheme or through a dedicated value head. Purpose-built questionnaires and improved uncertainty handling may also mitigate the observed difficulties with the `Neutral` class.

## 6 CONCLUSION

In this work, we explore individual opinion alignment by training persona-conditioned agents directly on known opinions and by framing answer generation as a structured reasoning task. Across Swiss candidates (*smartvote*), German parties (*Wahl-o-Mat*), and U.S. voters (ANES), `SFT+GRPO` generally outperforms all baselines. Performance remains uneven as `Neutral` stances remain challenging to predict, and accuracy varies significantly across the political spectrum. Our work provides a strong benchmark and baseline for systematic opinion modeling with LLMs, marking an early step toward the possibility of more representative, AI-driven forms of democracy.

## ETHICS STATEMENT

We acknowledge that the modeling of individual opinions in a time of rising mis- and disinformation is not exclusively beneficial, and may pose downstream risks. Precisely because of these risks and the technology's inevitable development, we believe responsible research is essential to investigate how individual opinion alignment can be improved. We use only anonymized survey data and public figures' positions to address privacy concerns. Our research reveals systematic performance disparities across ideological groups, with consistent underperformance on center and right-wing profiles, as well as difficulties modeling neutral positions. We commit to transparent reporting of these biases, as they represent significant system risks that could disenfranchise certain political viewpoints if deployed prematurely. We strongly caution against real-world applications until these fairness issues are resolved, as biased political modeling systems risk undermining democratic representation. Ultimately, we are convinced that with proper bias mitigation, opinion modeling can ultimately enable societally beneficial applications such as digital democracies, where models could help shape fairer and more popular policies.

## REPRODUCIBILITY STATEMENT

For reproducibility, we release our code and datasets:

```
https://github.com/ETH-DISCO/reasoning-boosts-llm-alignment
```

Training details, including hyperparameters, are given in Appendix F. Details for all PCA results are described in Appendix E.5.

### AUTHOR CONTRIBUTIONS

Frédéric Berdoz conceived and designed the project, led experimental planning, and defined its research direction. Yann Billeter contributed equally, leading technical development, including dataset collection, model training, evaluation, and analysis. Yann Vonlanthen contributed as an advisor, offering guidance on methodology and experimental design. Roger Wattenhofer supervised the project as principal investigator.

### ACKNOWLEDGMENTS

We thank Prof. Dr. Hans Gersbach for his valuable insights during the project.

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

# A  DATASET DETAILS

In the following, we provide more information on the datasets used in this work.

## A.1  SWISS PARTIES

Table 4 lists the Swiss parties included in the *smartvote* dataset, along with their political orientations and group assignments.

Table 4: The six major Swiss parties and their political positions. All parties' candidates answered the same 60 questions (48 train / 12 test).

| Party | Political Orientation | Group |
|---|---|---|
| SVP | Right-wing conservative | Right |
| SP | Center-left social democratic | Left |
| FDP | Center-right liberal | Right |
| The Center | Center/Christian democratic | Center |
| Green Party | Left-wing green | Left |
| GLP | Center-left green liberal | Center |

## A.2  WAHL-O-MAT

Table 5 lists the German parties included in the *Wahl-o-Mat* dataset, along with their political orientations, group assignments, as well as the train and test set statistics.

Table 5: Number of questions per party in the German dataset across all elections (2021–2025). The test sets contain 30 questions for all parties. `A → Yes, B → No, C → Neutral`.

| Party | Political Orientation | # Train Q. | Train. distr. Yes/Neutral/No | Test distr. Yes/Neutral/No | Group |
|---|---|---|---|---|---|
| CDU/CSU | Center-right Christian democratic | 646 | 52.32 / 39.16 / 8.51 | 60.0 / 36.67 / 3.33 | Right |
| SPD | Center-left social democratic | 760 | 55.92 / 32.63 / 11.45 | 63.33 / 33.33 / 3.33 | Center |
| Grüne | Center-left green | 722 | 52.08 / 37.40 / 10.53 | 73.33 / 20.0 / 6.67 | Left |
| FDP | Center-right liberal | 760 | 48.68 / 42.37 / 8.95 | 46.67 / 53.33 / 0.00 | Center |
| Die Linke | Left-wing socialist | 646 | 52.17 / 41.80 / 6.04 | 56.67 / 40.0 / 3.33 | Left |
| AfD | Far-right populist | 722 | 45.43 / 48.20 / 6.37 | 30.0 / 66.67 / 3.33 | Right |

## A.3  ANES

Table 6 provides the assignment of *ANES* ideologies to groups.

Table 6: Assignment of ANES ideology self-placement answers to groups.

| Ideology | Group |
|---|---|
| Extremely liberal | Left |
| Liberal | Left |
| Slightly liberal | Center |
| Moderate | Center |
| Slightly conservative | Center |
| Conservative | Right |
| Extremely conservative | Right |

## A.4  PROMPTS FOR SYNTHETIC DATASETS

The following prompt was used to generate the synthetic argument datasets.

```
You are a [politician / voter] in [COUNTRY]. You reply
in English only.  We present you with a political
issue or question, and you provide us with three to
five diverse yet concise arguments [for / against]
this issue or question.  [ISSUE]
```

For *smartvote* and *ANES*, COUNTRY was replaced with Switzerland and the USA, respectively. Similarly, politician was used for *smartvote*, while voter was used for *ANES*. For the biased datasets used in Section 4.8, the *smartvote* prompt was changed to the following:

```
You are a politician in Switzerland.  You reply in
English only.  We present you with a political issue
or question, and you provide us with a [progressive /
conservative] argument [for / against] this issue or
question.  [ISSUE]
```

## B    ADDITIONAL REASONING TRACES

In most cases, the reasoning traces are on-topic, align with the final answer, and are internally consistent. In Table 7, we present an example from a German party in the Center group (SPD). Here, the agent is also making use of information it acquired, most likely during the SFT stage on the training set.

Table 7: Example of successful reasoning. The reasoning trace and the final answer are consistent. The agent is also making use of additional information about the party position, most likely acquired during the SFT stage on the training set.

| |
|---|
| **Question:** Social programs should be maintained even at the cost of higher taxes |
| <reasoning> We are committed to social justice and therefore to strong social work and high social spending. For us, social policy always means more than mere benefit payments. We are committed to free childcare for all children up to the age of ten years and for senior citizens and nursing services that look after them with respect and dignity. </reasoning> <answer>A</answer> |

Agents sometimes resort to rather unconventional arguments. For instance, in the trace in Table 8, an agent trained on a left-wing politician from Switzerland suggested that non-proliferation could suffer if Switzerland were to build new nuclear power plants. While the argument and the reasoning are consistent, it is rather unlikely that this argument would be made.

Table 8: Example of a reasoning trace where the reasoning is formally correct, but the argument made is unlikely to be made in this context.

| |
|---|
| **Question:** Should the construction of new nuclear power plants in Switzerland be allowed again? |
| <reasoning > Switzerland has historically been committed to a policy of non-proliferation of nuclear weapons. Allowing new nuclear power plants could potentially compromise this policy, as there is a risk that nuclear materials or technology could be diverted or transferred to other countries with questionable intentions, potentially contributing to the development of nuclear weapons.</reasoning > <answer >B</answer > |

Similarly, in Table 9, the agent starts in the right direction for the unit it is trained to represent, but then switches topics to prostitution.

## C    ADDITIONAL FIGURES

Figure 8 provides per-party F1 scores, similar to Fig. 3.

Table 9: Example of a trace where the model goes off-topic. This is most likely prompted by the presence of the word "sex".

| |
|---|
| **Question:** The legalization of same-sex marriages is a good thing. |
| `<reasoning>` In the future, there should be no further debates on the equality of homo and heterosexual couples. There must also be further prostitution-free areas to protect the neighborhood from the influence of street prostitution. `</reasoning>`
`<answer>`A`</answer>` |

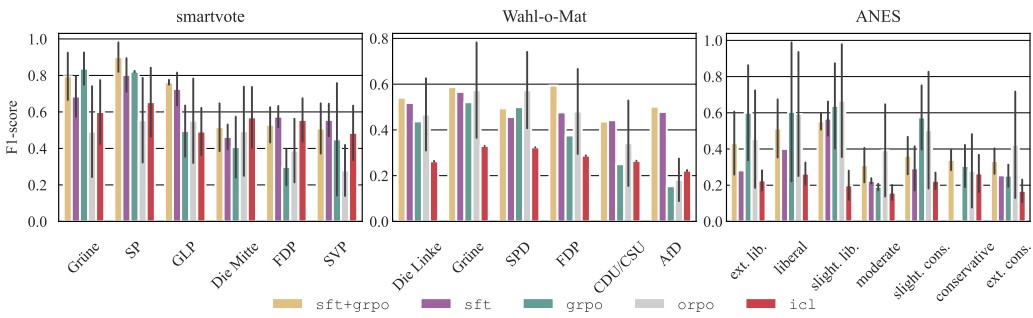

Figure 8: Same data as Fig. 3, broken down by party. The party-level data shows a similar trend, `SFT+GRPO` generally outperforms `SFT` and `GRPO`. F1 tends to be better on left and center-left parties, except the German center-right FDP.

## D    ANES RECODING SCHEMES

Questions that ask for (dis-)agreement on a Likert scale were left as is, and the answer options are mapped such that all agreeing options become "Yes" or "Agree", and all disagreeing options become "No" or "Disagree". Options like "Neither agree nor disagree" or "None of the above" are mapped to neutral. All other types of questions were rewritten to allow for this answer style. For instance, consider the following original ANES question:

```
When women demand equality these days, How often are
they actually seeking special favors?
1.  Always
2.  Most of the time
3.  About half the time
4.  Some of the time
5.  Never
```

This question can be rewritten as a statement like:

```
When women demand equality these days, they are
actually seeking special favors.
```

For recoding the answers, we consider two schemes:

- Conservative scheme: The original options 1 and 2 become `Yes`, option 3 becomes `Neutral`, and options 4 and 5 are mapped to `No`. Generally, options in the middle of the scale, like `moderately important`, `a moderate amount`, `about half of the time`, etc., are mapped to `Neutral` under this scheme.
- Aggressive scheme: Options 1 to 4 are mapped to `Yes` and only option 5 is mapped to `No`. Generally, only clearly negative options are mapped to `No` under this scheme. There is no `Neutral` option, unless the original options contain one.

The conservative scheme treats only clearly positive frequency responses as agreement, while the aggressive scheme treats any non-zero frequency as agreement.

## D.1 FURTHER EXAMPLES

The question

```
How often do Members of Congress change their votes
on legislation because someone donates money to their
campaign?
1.  Never
2.  Rarely
3.  A moderate amount of time
4.  Very often
5.  All the time
```

becomes

```
Do Members of Congress change their votes on
legislation because someone donates money to their
campaign?
```

with mappings $(4, 5) \rightarrow$ Yes , $(3) \rightarrow$ Neutral, $(1, 2) \rightarrow$ No using the conservative scheme, and $(2,3,4,5) \rightarrow$ Yes and $(1) \rightarrow$ No using the aggressive scheme. As an other example, the question

```
Does the increasing number of people of many different
races and ethnic groups in the United States make this
country a better place to live, a worse place to live,
or does it make no difference?
1.  Better
2.  Worse
3.  Makes no difference
```

becomes

```
Does the increasing number of people of many different
races and ethnic groups in the United States make this
country a better place?
```

with mappings $(1) \rightarrow$ Yes, $(2) \rightarrow$ No and $(3) \rightarrow$ Neutral in both recoding schemes.

## E  ADDITIONAL MATERIAL

The following sections provide additional material to contextualize the findings in the main text.

### E.1  ADDITIONAL RESULT TABLES

Table 10 presents accuracy scores across models and configurations. Similarly, in Table 11, we present accuracy scores without the Neutral class. In Table 12 and Table 13 , we report one-tailed Welch's t-tests (SFT+GRPO > baseline) and Cohen's $d$ effect sizes for the results in Table 3 (macro-F1) and Table 10 (mean accuracy), respectively.

### E.2  IMPACT OF RECODING ANES

Since the recoding is not unique and can potentially lead to information loss and semantic shifts, we investigate the impact of different recoding schemes on the political groups present in the *ANES* dataset. Details on the different schemes are provided in Appendix D. The key difference lies in how moderate responses are treated: the conservative scheme maps them to neutral/disagreement,

Table 10: **Mean accuracies** (%) with standard deviations, averaged over 8 stochastic decoding runs at temperature $T=1.0$. For each run, we first compute the per-unit accuracy (candidate/party/respondent) over that unit's test items and then average across units. The table reports the across-run mean $\pm$ s.d. Baselines: `random` (uniform; 50.0/33.3 on binary/ternary labels by construction) and `majority` (deterministic). Bold indicates the best method per base model and dataset, with significance reported in Table 13.

| Base model | Method | Dataset | | |
|---|---|---|---|---|
| | | *smartvote* | *WoM* | *ANES* |
| Untrained baselines | random | 50.00 | 33.33 | 33.33 |
| | majority | 37.43 | 27.44 | 22.98 |
| Llama 3.1 8B | icl | $63.91 \pm 3.79$ | $31.44 \pm 0.02$ | $44.39 \pm 1.43$ |
| | ORPO | $59.23 \pm 2.47$ | $57.48 \pm 5.97$ | $44.15 \pm 5.61$ |
| | SFT | $67.23 \pm 1.00$ | $68.25 \pm 4.30$ | $58.82 \pm 0.93$ |
| | GRPO | $66.44 \pm 0.86$ | $60.63 \pm 1.50$ | $52.35 \pm 1.35$ |
| | SFT+GRPO | $\mathbf{70.53 \pm 1.51}$ | $\mathbf{75.06 \pm 3.30}$ | $\mathbf{59.21 \pm 0.62}$ |
| Qwen3 8B | icl | $66.09 \pm 2.85$ | $48.67 \pm 0.73$ | $41.32 \pm 2.98$ |
| | ORPO | $39.02 \pm 2.04$ | $38.56 \pm 3.51$ | $34.07 \pm 2.08$ |
| | SFT | $65.18 \pm 1.91$ | $61.74 \pm 4.63$ | $47.01 \pm 1.79$ |
| | GRPO | $67.04 \pm 1.69$ | $53.19 \pm 2.33$ | $52.92 \pm 0.72$ |
| | SFT+GRPO | $\mathbf{71.27 \pm 2.83}$ | $\mathbf{71.16 \pm 1.60}$ | $\mathbf{55.14 \pm 0.41}$ |
| Magistral 24B | icl | $71.41 \pm 3.54$ | $44.64 \pm 0.67$ | $42.46 \pm 5.13$ |
| | ORPO | $35.93 \pm 3.66$ | $36.10 \pm 5.18$ | $31.32 \pm 2.23$ |
| | SFT | $70.83 \pm 1.58$ | $72.56 \pm 2.40$ | $52.80 \pm 0.94$ |
| | GRPO | $68.44 \pm 1.23$ | $72.05 \pm 2.99$ | $62.33 \pm 0.51$ |
| | SFT+GRPO | $\mathbf{73.92 \pm 1.89}$ | $\mathbf{75.10 \pm 2.80}$ | $\mathbf{62.33 \pm 0.78}$ |

Table 11: **Mean accuracies** (%) with standard deviations when `Neutral` is removed. Results are the averages over 8 stochastic decoding runs at temperature $T=1.0$. For each run, we first compute the per-unit accuracy (candidate/party/respondent) over that unit's test items and then average across units. The table reports the across-run mean $\pm$ s.d. Baselines: `random` (uniform; 50.0/33.3 on binary/ternary labels by construction) and `majority` (deterministic). Bold indicates the best method per base model and dataset (higher is better). *smartvote* results are unchanged from Table 10 because there are no neutral answers on *smartvote*.

| Base model | Method | Dataset | | |
|---|---|---|---|---|
| | | *smartvote* | *WoM* | *ANES* |
| Untrained baselines | random | 50.00 | 33.33 | 33.33 |
| | majority | 37.43 | 27.44 | 22.98 |
| Llama 3.1 8B | icl | $63.91 \pm 3.79$ | $31.03 \pm 0.19$ | $47.15 \pm 1.69$ |
| | ORPO | $59.23 \pm 2.47$ | $59.28 \pm 6.15$ | $44.15 \pm 5.61$ |
| | SFT | $67.23 \pm 0.10$ | $70.11 \pm 4.36$ | $62.47 \pm 0.92$ |
| | GRPO | $66.44 \pm 0.86$ | $62.93 \pm 1.54$ | $64.15 \pm 1.51$ |
| | SFT+GRPO | $\mathbf{70.53 \pm 1.51}$ | $\mathbf{78.19 \pm 3.38}$ | $\mathbf{71.35 \pm 0.80}$ |
| Qwen3 8B | icl | $66.09 \pm 2.85$ | $31.03 \pm 0.19$ | $19.11 \pm 2.93$ |
| | ORPO | $39.02 \pm 2.04$ | $37.98 \pm 3.29$ | $37.04 \pm 1.49$ |
| | SFT | $65.18 \pm 1.91$ | $63.62 \pm 4.74$ | $54.38 \pm 2.03$ |
| | GRPO | $67.04 \pm 1.69$ | $55.30 \pm 2.44$ | $65.38 \pm 0.91$ |
| | SFT+GRPO | $\mathbf{71.27 \pm 2.83}$ | $\mathbf{73.69 \pm 1.60}$ | $\mathbf{67.55 \pm 0.72}$ |
| Magistral 24B | icl | $66.09 \pm 2.85$ | $46.62 \pm 0.80$ | $46.27 \pm 5.94$ |
| | ORPO | $35.93 \pm 3.66$ | $36.39 \pm 5.38$ | $33.67 \pm 1.86$ |
| | SFT | $70.83 \pm 1.58$ | $74.71 \pm 2.71$ | $62.67 \pm 1.17$ |
| | GRPO | $68.44 \pm 1.23$ | $\mathbf{77.21 \pm 2.72}$ | $66.34 \pm 0.55$ |
| | SFT+GRPO | $\mathbf{73.92 \pm 1.89}$ | $74.71 \pm 2.71$ | $\mathbf{69.99 \pm 0.85}$ |

while the aggressive scheme treats most non-zero responses as agreement. We train Llama 3.1 8B on the two recoding schemes and find that mean F1-scores on the aggressive scheme are higher than on the conservative one ($45.33 \pm 1.37$ vs. $40.67 \pm 0.93$ reported in Table 3). The confusion matrices in Fig. 9 indicate that the improvement is mostly driven by a lower number of `Neutrals` in the aggressive scheme. These results highlight that recoding choices can substantially impact both model performance and the political diversity represented in simulated populations.

Table 12: **Statistical significance and effect sizes for `SFT+GRPO` improvements in macro-F1**. Results correspond to the experiments reported in Table 3. We perform one-tailed Welch's t-tests (`SFT+GRPO` > baseline) and report Cohen's $d$ effect sizes. Green: significant after Bonferroni correction ($p < 0.0014$ for 36 tests). Blue: significant without correction ($p < 0.05$). Red: not significant.

| Model | Baseline | P-values | | | Cohen's d | | |
|---|---|---|---|---|---|---|---|
| | | *smartvote* | *WoM* | *ANES* | *smartvote* | *WoM* | *ANES* |
| Llama 3.1 8B | icl | 0.0003 | 0.0000 | 0.0000 | 2.54 | 8.51 | 12.30 |
| | ORPO | 0.0000 | 0.0008 | 0.0093 | 9.29 | 1.99 | 1.50 |
| | SFT | 0.0013 | 0.0409 | 0.9991 | 1.90 | 0.94 | -1.95 |
| | GRPO | 0.0000 | 0.0000 | 0.0000 | 6.61 | 4.74 | 5.39 |
| Qwen3 8B | icl | 0.0140 | 0.0000 | 0.0000 | 1.23 | 16.72 | 11.75 |
| | ORPO | 0.0000 | 0.0000 | 0.0000 | 14.87 | 8.39 | 11.46 |
| | SFT | 0.0099 | 0.0004 | 0.0004 | 1.32 | 2.32 | 2.67 |
| | GRPO | 0.0037 | 0.0000 | 0.0000 | 1.63 | 10.50 | 8.70 |
| Magistral 24B | icl | 0.0102 | 0.0000 | 0.0000 | 1.36 | 11.91 | 11.90 |
| | ORPO | 0.0000 | 0.0000 | 0.0000 | 21.95 | 8.79 | 11.81 |
| | SFT | 0.0049 | 0.1842 | 0.0000 | 1.50 | 0.47 | 6.88 |
| | GRPO | 0.0000 | 0.0909 | 0.0046 | 4.90 | 0.70 | 1.51 |

Table 13: **Statistical significance and effect sizes for `SFT+GRPO` improvements in accuracy**. Results correspond to the experiments reported in Table 10. We perform one-tailed Welch's t-tests (`SFT+GRPO` > baseline) and report Cohen's $d$ effect sizes. Green: significant after Bonferroni correction ($p < 0.0014$ for 36 tests). Blue: significant without correction ($p < 0.05$). Red: not significant.

| Model | Baseline | P-values | | | Cohen's d | | |
|---|---|---|---|---|---|---|---|
| | | *smartvote* | *WoM* | *ANES* | *smartvote* | *WoM* | *ANES* |
| Llama 3.1 8B | icl | 0.0006 | 0.0000 | 0.0000 | 2.29 | 18.66 | 13.45 |
| | ORPO | 0.0000 | 0.0000 | 0.0001 | 5.52 | 3.64 | 3.77 |
| | SFT | 0.0001 | 0.0017 | 0.1714 | 2.58 | 1.78 | 0.49 |
| | GRPO | 0.0000 | 0.0000 | 0.0000 | 3.33 | 5.63 | 6.53 |
| Qwen3 8B | icl | 0.0013 | 0.0000 | 0.0000 | 1.82 | 18.09 | 6.50 |
| | ORPO | 0.0000 | 0.0000 | 0.0000 | 13.07 | 11.95 | 14.06 |
| | SFT | 0.0001 | 0.0002 | 0.0000 | 2.52 | 2.72 | 6.26 |
| | GRPO | 0.0019 | 0.0000 | 0.0000 | 1.81 | 8.99 | 3.79 |
| Magistral 24B | icl | 0.0527 | 0.0000 | 0.0000 | 0.88 | 14.96 | 5.42 |
| | ORPO | 0.0000 | 0.0000 | 0.0000 | 13.04 | 9.37 | 18.56 |
| | SFT | 0.0017 | 0.0361 | 0.0000 | 1.77 | 0.97 | 11.03 |
| | GRPO | 0.0000 | 0.0269 | 0.5000 | 3.44 | 1.05 | 0.00 |

### E.3 REGRESSION TABLES

Table 14 provides the regression table for the regression shown in Fig. 4. Similarly, Table 15 provides the regression table for the corresponding analysis shown in Fig. 10.

Table 14: Regression table for the analysis presented in Fig. 4 and Section 4.6: Model performance vs. `Neutral` base rate.

| Metric | N | Intercept ($\alpha$) | Slope ($\beta$) | Slope SE | Slope 95% CI |
|---|---|---|---|---|---|
| F1 Score (Macro) | 21 | 0.5517 | -0.6741 | 0.2091 | [-1.3039, -0.4084] |
| Accuracy | 21 | 0.7803 | -0.8562 | 0.2139 | [-1.1117, -0.2364] |

| Metric | r | $R^2$ | p-value | RMSE | |
|---|---|---|---|---|---|
| F1 Score (Macro) | -0.5946 | 0.3535 | 0.004472 | 0.1250 | |
| Accuracy | -0.6763 | 0.4574 | $< 0.001$ | 0.1279 | |

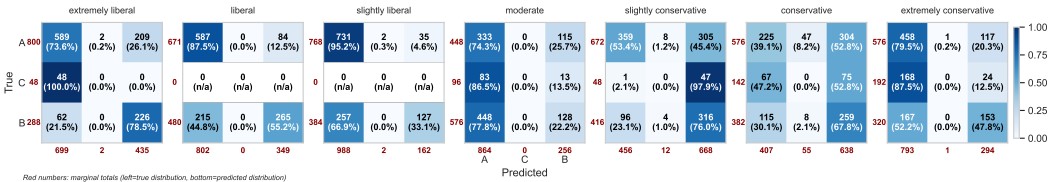

Figure 9: Confusion matrices for Llama 3.1 8B (`SFT+GRPO`) trained on *ANES* with a more aggressive recoding scheme. `A` → `Yes`, `B` → `No`, `C` → `Neutral`.

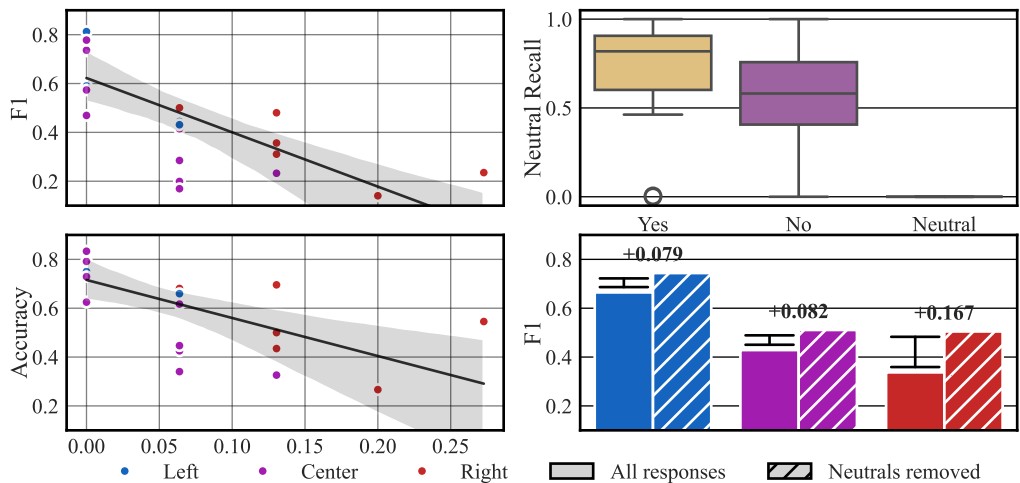

Figure 10: F1 score (A) and accuracy (B) vs. `Neutral` response rate. Predictive performance decreases significantly as `Neutral` response rate increases. Comparing the results presented here to those in Fig. 4, we observe the same trend regardless of the coding scheme. (D): Performance in terms of F1 score on ANES before (left) and after (right) removing `Neutral` instances. All groups improve, but Center only marginally. The gap between Center and Right closes. Performance in predicting `Neutral` decreased (C). Results were obtained for Llama 3.1 8B using the `SFT+GRPO` configuration.

### E.4    IMPACT OF THE NUMBER OF TRAINING QUESTIONS

We ablate the number of questions in the *WoM* training set for the `SFT+GRPO` configuration. The results for Llama 3.1 8B are presented in Fig. 11. Increasing the number of training questions is beneficial for all parties, except the center-right FDP party. The FDP's performance may be explained by a shift in the party's positions due to dwindling voter numbers between 2021 and 2025. The largest increase in accuracy is observed for the far-right AfD, which we attribute to the fact that their positions are furthest from the LLM's position, and SFT thus has an aligning or "corrective" effect which becomes stronger as more data is added.

### E.5    PCA DETAILS

To obtain the results shown in Fig. 2 and Fig. 12, we compute the PCA as follows: We use the complete *smartvote* dataset for the 2023 national elections across all major parties and all candidates. We convert the original 4-point Likert scale answers to yes-no, as we did for the *smartvote* subset studied in the remainder of the paper. `Yes` answers are subsequently mapped to 1, and `No` answers to 0. We then compute the PCA of the resulting matrix, where the rows correspond to candidates, and the columns correspond to questions.

Table 15: Regression table for the analysis presented in Fig. 10: Model performance vs. `Neutral` base rate.

| Metric | N | Intercept ($\alpha$) | Slope ($\beta$) | Slope SE | Slope 95% CI |
|---|---|---|---|---|---|
| F1 Score (Macro) | 21 | 0.6225 | $-2.2231$ | 0.5049 | $[-3.2799, -1.1662]$ |
| Accuracy | 21 | 0.7157 | $-1.5571$ | 0.4496 | $[-2.4981, -0.6161]$ |

| Metric | r | $R^2$ | p-value | RMSE |
|---|---|---|---|---|
| F1 Score (Macro) | $-0.7106$ | 0.5050 | $< 0.001$ | 0.1608 |
| Accuracy | $-0.6221$ | 0.3870 | 0.002603 | 0.1431 |

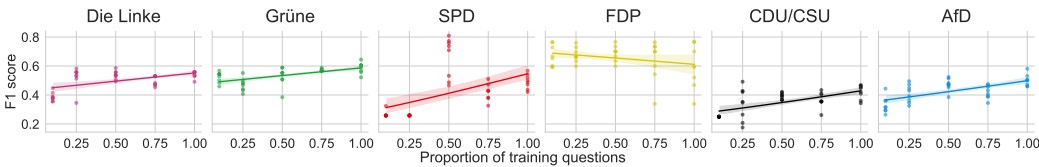

Figure 11: Impact of the number of questions in the training set. Shown are accuracies for 10%, 25%, 50%, 75%, and 100% of available training questions per party on the *WoM* dataset. Overall, F1 increases as the number of training questions increases. The largest increase is observed for the far-right AfD, while the FDP demonstrates a counterexample.

For agents, we replace the ground-truth test set answers with the agents' predictions, and project the resulting vectors onto the first two principal components. Similarly, for the inverted positions shown in Fig. 12, we invert all ground-truth answers and project the resulting vectors.

## E.6 INVERSION

Figure 12 shows candidate positions before and after flipping all answer options. As can be seen, this operation roughly corresponds to a rotation by $180°$ about the origin.

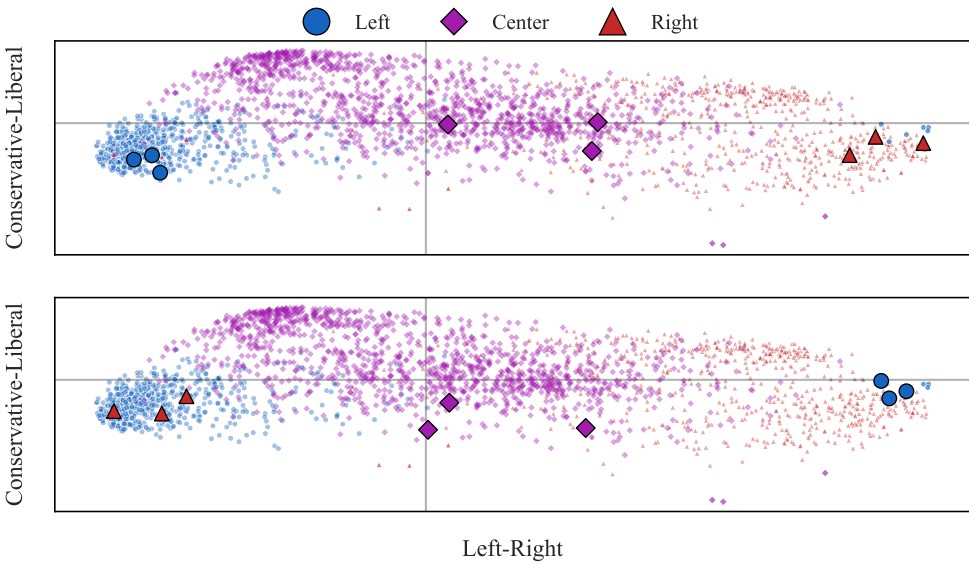

Figure 12: Inverting all answers in the dataset roughly corresponds to a rotation by $180°$ about the origin. Top: Candidates before inversion. Bottom: Candidates after inversion.

## F  TRAINING DETAILS

In the following, we provide information on training settings and hyperparameters.

**System prompt.**   We used the following system prompt

```
You are a digital twin of a [NATIONALITY] voter.  You
are asked a policy issue or question.  You must reason
and then answer the question as if you were the voter.
You reason and answer in English.  Your final answer
must be one of the answer options ("A" for "Yes", "B"
for "No", "C" for "Neutral").  You must respond in the
following format:
<reasoning>
Your reasoning goes here.
</reasoning>
<answer>
Final answer, one of
A) Yes,
B) No,
C) Neutral.
</answer>
```

where we replaced `[NATIONALITY]` with `Swiss` for *smartvote*, `German` for *Wahl-o-Mat*, and `US` for *ANES*.

**LoRA settings.**   We use the following LoRA settings for all models and methods. No hyperparameter optimization was performed. $r = 32, \alpha = 32$, target modules: `q_proj`, `k_proj`, `v_proj`, `o_proj`, `gate_proj`, `up_proj`, `down_proj`.

**ORPO.**   We use the same hyperparameters as Stammbach et al. (2024) and the Hugging Face implementation of ORPO.

**SFT.**   We train for 800 steps using the Adam optimizer with a batch size of 8, learning rate $5e - 5$, a cosine learning rate scheduler, and we fix the maximum gradient norm at 1.0. The first 80 steps were used as a warm-up period. For reasoning-pretrained models, we change the `<reasoning></reasoning>` tags to their respective reasoning templates when fine-tuning. We used the Hugging Face implementation of SFT, and all other hyperparameters were left at the default values.

**GRPO.**   We train for 800 steps using the Adam optimizer with a batch size of 8, learning rate $5 \times 10^{-6}$, a cosine learning rate scheduler, and we fix the maximum gradient norm at 1.0. We use the Adam optimizer with $\beta_1 = 0.9$ and $\beta_2 = 0.99$, weight decay with 0.1. The first 80 steps were used as a warm-up period. The GRPO group size was 8. The GRPO $\beta$ parameter was set to 0. The temperature was set at 1.0. For reasoning-pretrained models, we change the `<reasoning></reasoning>` tags to their respective reasoning templates in the format of reward functions. We used the Hugging Face implementation of GRPO, and all other hyperparameters were left at the default values.

**SFT+GRPO.**   For this configuration, we used the same settings as for the individual `SFT` and `GRPO` configurations.

