# OpenReview forum: "Reasoning Boosts Opinion Alignment in LLMs"
_ICLR.cc/2026/Conference — ICLR 2026 Poster_

### Official Review · Reviewer_zvbd · 2025-10-26

**Soundness:** 2
**Presentation:** 3
**Contribution:** 2
**Rating:** 4
**Confidence:** 3

**Summary:**

The authors present a novel approach toward aligning LLMs with individual opinions in a political context, which could enable downstream applications such as more representative digital democracies. In contrast to prior work that focuses on in-context learning, they use reinforcement learning to train models to align to the political opinions of individual candidates, finding gains in opinion alignment when compared to certain baselines. They also perform a detailed analysis of results and find that models are more easily aligned to left-wing political groups, but that trained agents tend to lean more centrist.

**Strengths:**

- The authors identify a key problem in opinion alignment that current prompting-based approaches fail to adequately solve, and that could enable highly prosocial applications if solved, and propose a novel solution.
- I found the political ideology analysis (sections 4.5, 4.7, 4.8) really interesting, especially the experiment with flipped labels in 4.8 that showed that left-leaning opinion profiles are easier to learn than synthetic left-leaning profiles constructed by inverting right-leaning opinion profiles. The findings that right and center positions are harder to learn than left ones, and that fine-tuning tends to push agents to the center, make interesting additions to the dialogue around opinion modeling with LLMs.

**Weaknesses:**

- The novel contribution of this work (using RL to train models to reason for opinion alignment) doesn’t seem to yield much improvement over SFT on seen opinions. In Table 3, SFT+GRPO is only a statistically significant improvement over SFT in three of the 12 base model - dataset combinations (Qwen3 8B + WoM, Qwen3 8B + ANES, Magistral 24B + ANES). The authors note that this relates to dataset scale, but SFT+GRPO doesn’t seem to consistently improve over just SFT even on the largest WoM dataset. Given the difficulty of collecting individual opinions, it may be difficult to apply the proposed method to new users.
- Given the amount of prior work on opinion modeling with LLMs in survey environments, I found “Our work lays the groundwork and introduces the first benchmark for systematic opinion modeling with LLMs” to be a bit of an overclaim.

**Questions:**

- How does opinion alignment scale with the number of questions personalized models are trained on (SFT, GRPO) or prompted with (ICL)? Analyzing the scaling behavior could help us understand what kind of approaches are most likely to work with current datasets, or inform the collection of future datasets.
- Why do you think your trained agents are generally more centrist and conservative than the opinions they are trained on (Figure 2), and how does this vary across different training/prompting settings?

---

> ### Author Response · Authors · 2025-11-23
>
> We thank the reviewer for their valuable feedback and address each concern below.
>
> > The novel contribution of this work (using RL to train models to reason for opinion alignment) doesn’t seem to yield much improvement over SFT on seen opinions. In Table 3, SFT+GRPO is only a statistically significant improvement over SFT in three of the 12 base model - dataset combinations (Qwen3 8B + WoM, Qwen3 8B + ANES, Magistral 24B + ANES). The authors note that this relates to dataset scale, but SFT+GRPO doesn’t seem to consistently improve over just SFT even on the largest WoM dataset. Given the difficulty of collecting individual opinions, it may be difficult to apply the proposed method to new users.
>
> We appreciate the reviewer’s concern, which gives us the opportunity to clarify the significance of our contributions and results. First, in addition to a new method (GRPO for opinion alignment), our work provides a curated benchmark dataset to enable fair and reproducible comparison in opinion modeling. Second, regarding improvements over SFT, we have added a one-tailed Welch’s t-tests (testing \texttt{SFT+GRPO} $>$ baselines) and find that almost all results are statistically significant, demonstrating that the method reliably enhances opinion alignment. We report these new results in Appendix E.1 (Table 12). Third, the main tables report macro-F1 scores, which demand high precision and recall across all classes and can make improvements appear smaller. In contrast, overall accuracies reported in Appendix E.1 (Table 11) show that SFT+GRPO consistently outperforms all baselines across models and datasets. Statistical significance analyses on these accuracies can be found in Table 13.
>
> > Given the amount of prior work on opinion modeling with LLMs in survey environments, I found “Our work lays the groundwork and introduces the first benchmark for systematic opinion modeling with LLMs” to be a bit of an overclaim.
>
> We appreciate the reviewer’s feedback and have revised the sentence to more accurately reflect the contribution of our work: “Our work provides a strong benchmark and baseline for systematic opinion modeling with LLMs.”.
>
> > How does opinion alignment scale with the number of questions personalized models are trained on (SFT, GRPO) or prompted with (ICL)? Analyzing the scaling behavior could help us understand what kind of approaches are most likely to work with current datasets, or inform the collection of future datasets.
>
> As seen in the ablation shown in Appendix E.4, the number of questions generally improves accuracy, though the effect size varies by ideology. Since accuracy is bounded while the number of questions is not, the precise scaling behavior is inherently difficult to characterize and compare across methods.
>
> > Why do you think your trained agents are generally more centrist and conservative than the opinions they are trained on (Figure 2), and how does this vary across different training/prompting settings?
>
> Our newly added Section 4.8 (“Impact of SFT data”) with Figures 5 and 6 addresses this question.
>
> First, our results show that ideological bias in the SFT data mainly harms the underrepresented group rather than consistently pulling models toward the overrepresented ideology. For example, Figure 5 shows that progressive-biased data disproportionately reduces performance on Right-leaning respondents, while conservative-biased data primarily harms Left-leaning respondents.
>
> Second, Figure 6 examines how these biases affect the agents’ PCA positions. Here again, the largest positional shifts occur for the underrepresented ideology. Notably, the center-right shift observed under both conservative-biased and default SFT data suggests that this effect is largely driven by the base model’s own alignment rather than the SFT data itself. Our default SFT data contained no explicit political framing, so the similarity between these two conditions points to the base-model alignment as the likely source. Indeed, modern LLMs are often trained to avoid strong political positions, which may encourage neutral or centrist answers.
>
> We hope the new experiments and explanations strengthen the reviewer’s overall assessment, and we remain available to provide any additional details or analyses.

---

> > ### Comment · Reviewer_zvbd · 2025-11-26
> >
> > Thanks for the response; I've raised my score from 4 to 6.

---

### Official Review · Reviewer_mY5k · 2025-10-29

**Soundness:** 3
**Presentation:** 3
**Contribution:** 2
**Rating:** 4
**Confidence:** 3

**Summary:**

This paper looks at whether reasoning helps align LLMs with human political opinions. The authors fine-tune LLMs on political survey data from the US, Germany, and Switzerland, and then train them further using Group Relative Policy Optimization (GRPO) — a reinforcement learning method that encourages structured reasoning before giving an answer. The goal is to make models answer policy questions like real individuals or parties would. Experiments show that reasoning-based training (SFT+GRPO) outperforms prompt-only and SFT-only baselines, improving F1 scores across all datasets. Overall, the paper argues that reasoning improves opinion alignment but isn’t enough to eliminate political bias entirely.

**Strengths:**

1. The paper studies a very relevant question — how reasoning affects opinion alignment, which feels timely and important.

2. The method is clear and builds nicely on recent RL reasoning techniques.

3. The experiments are well-organized and tested across multiple real-world political datasets.

4. The analysis is thoughtful, especially the discussion of ideological bias and neutral-class difficulty.

**Weaknesses:**

1. The clarity of paper should be further refined, and there are several typos in the content.

2. It’s not entirely clear whether GRPO genuinely improves reasoning or just fits survey patterns better.

3. The paper shows biases but doesn’t really explain their sources or propose fixes.

**Questions:**

1. Could you please provide a more detailed explanation of Figure 4, especially the panels on the right? I would like to understand what those plots specifically represent and what the error bars indicate in this context.

2. Can the authors show evidence that GRPO improves reasoning quality rather than just accuracy?

3. Did you analyze why right-leaning or neutral profiles are harder to learn — is it data imbalance or model bias?

---

> ### Author Response · Authors · 2025-11-23
>
> We thank the reviewer for their valuable feedback and address each concern below.
>
> > The clarity of paper should be further refined, and there are several typos in the content.
>
> We apologize for the typos. We have corrected those we identified and improved the wording in several sections for greater clarity. If there are specific parts the reviewer finds unclear, we would be grateful for pointers so we can address them.
>
> > The paper shows biases but doesn’t really explain their sources or propose fixes.
>
> We agree that identifying the sources of bias and developing mitigation strategies are important open challenges. Our work, however, focuses on analyzing GRPO and empirically demonstrating its effect on opinion alignment. As highlighted in recent surveys [1,2,3], fully understanding the origins of biases in LLMs is a complex task, and we do not have access to all details of the pre- and post-training processes, preventing a thorough analysis of their sources.
>
> > Could you please provide a more detailed explanation of Figure 4, especially the panels on the right? I would like to understand what those plots specifically represent and what the error bars indicate in this context
>
> - **Left column**: Regression of performance (F1 and accuracy) on neutral base rate. Each point shows ANES respondents’ performance versus their neutral answer rate, colored by political alignment. Right-leaning respondents tend to have higher neutral rates and lower performance, while left-leaning and centrist respondents have lower neutral rates and higher performance, showing a negative correlation.
> - **Top right**: Recall averaged over all respondents, grouped by answer option. Error bars indicate variation across respondents. Recall is highest for “yes” and lowest for “neutral.”
> - **Bottom right**: F1 before and after removing Neutral answers. Performance increases for all groups, though the smaller gain for the Right suggests neutral answers are not the sole cause of their underperformance.
>
> **Takeaways**: Neutral responses reduce performance unevenly across ideological groups. Even after adjustment, right-leaning respondents remain harder to model, indicating additional underlying factors.
>
> > It’s not entirely clear whether GRPO genuinely improves reasoning or just fits survey patterns better.
> > Can the authors show evidence that GRPO improves reasoning quality rather than just accuracy?
>
> We would like to clarify that our claim is not that GRPO directly improves reasoning quality. Rather, we aim to show that training LLMs to reason via GRPO leads to better overall opinion alignment (for survey questions). Table 3 provides quantitative evidence for this improvement. Regarding reasoning quality specifically, we include illustrative before/after reasoning traces in Table 1 and App. B, which provide qualitative evidence of how the model’s internal deliberation changes after GRPO.
>
> > Did you analyze why right-leaning or neutral profiles are harder to learn — is it data imbalance or model bias?
>
> We thank the reviewer for this interesting question. Neutral responses often reflect heterogeneous motivations (uncertainty, social desirability, or strategic behavior) rather than genuine neutrality, making them difficult to interpret. Learning Neutral answers is therefore challenging for LLMs, especially with reasoning-based models, and might reflect a limitation of the data collection process. Regarding the difficulty of modeling right-wing (and to some extent centrist) positions, several factors may contribute, though no definitive answer exists. Data imbalance is unlikely, as “Yes” and “No” answers are roughly even across groups. Model bias likely plays a role, as LLMs often exhibit left-liberal tendencies in multiple-choice assessments [4]. Post-training alignment may also reduce the ability to produce strongly ideological content. Additionally, ideological coherence differs across groups: Left-leaning respondents show less intra-group variation, making patterns easier to learn, while the Center and Right exhibit more diversity. Figure 7 provides strong evidence for this behavior. Finally, measurement effects in surveys may differentially affect political ideologies, representing another key unknown factor.
>
> We hope these additions help clarify the contribution and assist the reviewer in their overall assessment of our work.
>
> [1] I. O. Gallegos et al., _Bias and Fairness in Large Language Models: A Survey_, in Computational Linguistics, 2024.
>
> [2] Y. Guo et al., _Bias in Large Language Models: Origin, Evaluation, and Mitigation_, arXiv:2411.10915, 2024.
>
> [3] R. Ranjan et al., _A Comprehensive Survey of Bias in LLMs: Current Landscape and Future Directions_, arXiv:2409.16430, 2024.
>
> [4] Rettenbergeret al., _Assessing political bias in large language models_. Journal of Computational Social Science, 2025.

---

### Official Review · Reviewer_chku · 2025-10-30

**Soundness:** 3
**Presentation:** 3
**Contribution:** 3
**Rating:** 6
**Confidence:** 3

**Summary:**

This paper investigates whether reasoning-based approaches can improve LLMs' ability to model individual political opinions. The authors train models using reinforcement learning (specifically GRPO - Group Relative Policy Optimization) to generate reasoning traces before providing political stances, using survey data from three political contexts: Swiss candidates (smartvote), German parties (Wahl-O-Mat), and U.S. voters (ANES 2020). The key contribution is showing that structured reasoning through SFT+GRPO outperforms baseline approaches, though significant challenges remain with neutral stances and right-leaning political positions. The work establishes a benchmark for opinion modeling and releases datasets for future research.

**Strengths:**

1. First systematic study of using RL-based reasoning for individual-level political opinion modeling, moving beyond demographic-based approaches
2. Three datasets from different political systems (US, Germany, Switzerland) provide robust cross-cultural validation
3. The paper identifies a gap between demographic-prompted political simulation (Santurkar et al., 2023; Argyle et al., 2023) and individual-level agents that must stay consistent with a known survey profile.

**Weaknesses:**

1. Training one model per individual is computationally prohibitive for real-world deployment
2. The method still trains one model per persona, which is expensive and does not scale to population-level simulation; the paper acknowledges this, but the main method remains hard to apply in real digital-democracy settings where thousands of agents are needed.

**Questions:**

The current pipeline does SFT on synthetic arguments generated by Llama 3.1 70B. How sensitive is the final SFT+GRPO model to the style of these synthetic arguments? For example, if the initial CoT is written in a more partisan tone, will GRPO converge to a different political center?

---

> ### Author Response · Authors · 2025-11-23
>
> We thank the reviewer for their valuable feedback and address each concern in the following.
>
> > Training one model per individual is computationally prohibitive for real-world deployment
> > The method still trains one model per persona, which is expensive and does not scale to population-level simulation; the paper acknowledges this, but the main method remains hard to apply in real digital-democracy settings where thousands of agents are needed.
>
> We acknowledge that training one model per persona is more expensive than using a shared model (at least in terms of memory), and we agree that exploring parameter-sharing techniques is an important direction for future work. However, alternatives are not necessarily cheaper: any method that learns persona embeddings through optimization (rather than prompting) would also require backpropagation and therefore may incur large or even comparable computational costs. In addition, per-persona fine-tuning avoids opinion leakage between individuals, which is a non-trivial risk in shared-parameter approaches. Finally, because our method is post-training on an existing LLM, the computational cost is modest (e.g., Magistral-24B required ~5 GPU hours per model in our experiments).
>
> > The current pipeline does SFT on synthetic arguments generated by Llama 3.1 70B. How sensitive is the final SFT+GRPO model to the style of these synthetic arguments? For example, if the initial CoT is written in a more partisan tone, will GRPO converge to a different political center?
>
> We thank the reviewer for this observation, which prompted an interesting additional analysis. We added Section 4.8 and Figures 5 and 6 to investigate the impact of ideologically biased SFT data. Figure 5 shows that bias primarily harms the underrepresented perspective rather than systematically benefiting the overrepresented one: progressive bias strongly impairs Right-leaning respondents and inconsistently affects Left-leaning respondents, while conservative bias impairs Left-leaning respondents with limited gains for Right-leaning respondents. Figure 6 analyzes PCA positions, showing similar patterns. The positional shifts under conservative-biased and default SFT data are highly similar, suggesting the centerward shift is mainly driven by the agent’s base model rather than SFT bias (our default SFT dataset did not include explicit political framing).
>
> We hope these improvements will further reinforce the reviewer’s positive assessment and are happy to respond to any further inquiries.

---

> > ### Comment · Reviewer_chku · 2025-11-26
> >
> > I thank the authors for their efforts to address the raised points. My concerns have been resolved, and I keep my original score.

---

### Official Review · Reviewer_ywJy · 2025-11-02

**Soundness:** 3
**Presentation:** 3
**Contribution:** 3
**Rating:** 6
**Confidence:** 3

**Summary:**

This paper explores whether explicit reasoning improves *individual-level* opinion alignment in LLMs. The task is to predict a persona’s stance $(y\in{\text{Yes, Neutral, No}})$ on survey questions (q), given the persona’s past answers. The model outputs `<reasoning>...</reasoning><answer>...</answer>` and is trained with GRPO (group-relative PPO) using a composite reward for format correctness, rationale length, and answer agreement. An optional SFT warm-start with synthetic rationales is applied before GRPO. Experiments span smartvote (CH), Wahl-o-Mat (DE), and ANES (US), showing that SFT+GRPO improves macro-F1 (e.g., 70.7 on smartvote) and that Neutral responses and right-leaning personas are harder to predict. Code and data are released.

**Strengths:**

* **Clear, reproducible method and reward design.** The structured format and composite reward are explicit, with training hyperparameters (LoRA, steps, schedulers) fully reported.
* **Multiple datasets and ideologies.** The paper evaluates on smartvote (binary), Wahl-o-Mat (ternary), and ANES (ternary), with clear unit definitions and splits.
* **Variance reporting across 8 stochastic runs.** Results include mean ± s.d., improving robustness.
* **Insightful analyses of failure modes and class imbalance.** Figures analyzing ideology and Neutral class are carefully interpreted; removing Neutral boosts F1 but preserves ideological gaps.
* **Resource release.** Code and datasets are linked, supporting reproducibility and follow-up work.

**Weaknesses:**

1. **Missing strong baselines for political alignment.** The paper omits comparisons against recent specialized methods that align LLMs to political viewpoints using supervised preference optimization or domain corpora (e.g., Stammbach et al. 2024).

2. **Limited persona-conditioning alternatives.** The method trains *one model per persona*, not comparing to shared-parameter persona embedding methods.

3. **Synthetic SFT rationales risk leakage/priors.** Synthetic arguments generated by a large LLM may encode political priors.

4. **Scope of SOTA claim is limited.** No head-to-head with specialized political fine-tuning models.

5. **Evaluation depth on ANES is modest.** Only 21 respondents modeled, limiting statistical power.

6. **Fairness and calibration analyses are limited.** The paper lacks per-persona calibration or reliability metrics.

**Questions:**

1. Please compare against at least one specialized political-alignment baseline (e.g., Stammbach et al. 2024) and/or ORPO-style preference optimization.
2. Can you provide a *shared model* baseline that conditions on persona embeddings instead of per-persona training?
3. How sensitive are results to the synthetic SFT data source?
4. Can you expand ANES to more respondents and report per-respondent confidence intervals?
5. How did you prevent test leakage from SFT synthetic data?
6. Could you include calibration metrics (ECE, coverage-accuracy)?
7. What is the compute footprint per persona for GRPO training?

---

> ### Author Response · Authors · 2025-11-23
>
> We thank the reviewer for their valuable feedback and address each concern in the following.
>
> > **Missing strong baselines for political alignment.** The paper omits comparisons against recent specialized methods that align LLMs to political viewpoints using supervised preference optimization or domain corpora (e.g., Stammbach et al. 2024).
> > Please compare against at least one specialized political-alignment baseline (e.g., Stammbach et al. 2024) and/or ORPO-style preference optimization.
>
> We have added ORPO results to Table 3, using the same hyperparameters as Stammbach et al. (2024). ORPO's poor performance further supports the view that traditional alignment methods are not suited for aligning the political opinions of LLMs. While Stammbach et al. (2024) also use ORPO, they train separate models by party, which prevents adaptation to individual opinions. Additionally, their focus is primarily on text generation, whereas our work emphasizes voting simulation, making direct comparison with their trained models infeasible.
>
> > **Limited persona-conditioning alternatives.** The method trains one model per persona, not comparing to shared-parameter persona embedding methods.
> > Can you provide a shared model baseline that conditions on persona embeddings instead of per-persona training?
>
> We note that the ICL baseline already constitutes a shared-parameter persona-embedding method. The conditioning signal consists of the set of known answers from the persona, provided as context to the LLM. The KV cache of these Q/A pairs can thus be interpreted as the persona embedding. Is the reviewer aware of a shared-parameter technique that would provide additional expressive power beyond ICL?
>
> > **Synthetic SFT rationales risk leakage/priors.** Synthetic arguments generated by a large LLM may encode political priors.
> > How sensitive are results to the synthetic SFT data source?
>
> We have added a new section (4.8 Impact of SFT data) with two new figures (Figures 5 and 6) to the paper where we investigate the impact of differently biased SFT data on the performance of the agents. Our results show that ideological bias in SFT data primarily harms the underrepresented group rather than systematically improving the overrepresented one. Specifically, a progressive bias strongly impairs Right-leaning / conservative respondents, while showing inconsistent effects on Left-leaning / liberal respondents. Similarly, a conservative bias impairs Left-leaning / liberal respondents, with limited improvements for Right-leaning / conservative respondents. This suggests that a poorly chosen prior can degrade performance for underrepresented groups. As our newly added results also show, the SFT data source explains, at least in part, the observed shift to the center in the PCA space (c.f. the newly added Figure 6).
>
> > **Scope of SOTA claim is limited.** No head-to-head with specialized political fine-tuning models.
>
> We clarify that the main claim of our paper, as stated in the abstract and introduction, is that reasoning (i.e., GRPO) improves the opinion alignment of LLMs compared to traditional alignment baselines (ICL, SFT, ORPO). If the reviewer feels that we overstate our contribution, we would be happy to revise the manuscript to more accurately reflect it. Could the reviewer clarify which specialized opinion-modeling systems they have in mind (specifically those capable of conditioning on any persona using only survey responses)?
>
> > Evaluation depth on ANES is modest. Only 21 respondents modeled, limiting statistical power.
> > Can you expand ANES to more respondents and report per-respondent confidence intervals?
>
> We are currently training additional models to expand ANES coverage and will include results shortly.
>
> > How did you prevent test leakage from SFT synthetic data?
>
> The SFT data was generated without any knowledge of the the survey answers, and using the same prompts for each persona. Therefore, we do not believe that direct test leakage is possible.
>
> > **Fairness and calibration analyses are limited.** The paper lacks per-persona calibration or reliability metrics.
> > Could you include calibration metrics (ECE, coverage-accuracy)?
>
> We do not report standard calibration metrics because our models do not produce informative probabilities: after the reasoning trace, the final answer score is essentially 1.0. This is expected: GRPO treats opinion prediction as a reasoning task rather than a probabilistic one, making calibration inappropriate while still improving transparency through explicit reasoning steps.
>
> > What is the compute footprint per persona for GRPO training?
>
> For Llama 3.1 8B and Qwen3 8B, the SFT+GRPO setting takes around 4 hours on a single NVIDIA RTX A6000 GPU. For Magistral 24B, the same configuration completes in roughly 5 hours on the same hardware.
>
> We hope the added experiments and improvements further strengthen the reviewer’s positive assessment. We remain available to address any additional questions.

---

### Meta-Review · Area_Chair_rnjD · 2026-01-05

**Summary:**

This paper investigates whether explicit reasoning improves individual-level opinion alignment in large language models. They propose a reinforcement-learning-based post-training framework using Group Relative Policy Optimization (GRPO) to induce structured reasoning traces and persona-consistent political opinions.


Reviewers acknowledged contributions


- Provides a systematic empirical evaluation across three political survey datasets covering U.S., German, and Swiss contexts.

- Shows that reasoning-based training generally improves opinion alignment compared to non-reasoning baselines.

- Identifies the persistent difficulty of modeling neutral and right-leaning opinion profiles.


Reviewer concerns


- Perceived overclaiming regarding state-of-the-art performance and “first benchmark” assertions.

- Scalability limitations due to the need to train a separate model per individual persona.

- Questions about the practical significance of the observed performance gains.

**Reviewer Concerns:**

Author responses during discussion

- Added comparisons with ORPO, which performed worse and thereby supported the effectiveness of the proposed approach.
- Provided statistical validation (e.g., t-tests) to substantiate the significance of performance improvements.
- Clarified and justified the computational cost of the method.
- Revised the manuscript to tone down statements that could be interpreted as overclaims.

Reviewers’ concerns were largely addressed through additional experiments and clarifications.

**Reviewer Scores:**

The contribution is viewed as solid but incremental rather than highly impactful. Review scores are expected to be maintained or slightly increased, with consensus evaluations likely around a score of 6.

---

### Decision · Program_Chairs · 2026-01-26

Accept (Poster)